# Identification and Functional Validation of Two Novel Antioxidant Peptides in Saffron

**DOI:** 10.3390/antiox13030378

**Published:** 2024-03-20

**Authors:** Yiyang Long, Han Tao, Shiyu Wang, Bingcong Xing, Zhineng Wang, Kexin Liu, Qingsong Shao, Fei Gao

**Affiliations:** Zhejiang Provincial Key Laboratory of Resources Protection and Innovation of Traditional Chinese Medicine, Zhejiang A&F University, Hangzhou 311300, China; 2021102062006@stu.zafu.edu.cn (Y.L.); taohan@zafu.edu.cn (H.T.); 2023613032021@stu.zafu.edu.cn (S.W.); bcxing@zafu.edu.cn (B.X.); 20230073@zafu.edu.cn (Z.W.); 20230205@zafu.edu.cn (K.L.)

**Keywords:** *Crocus sativus* L., antioxidant peptide, cellular antioxidant activity, plant bioactive compounds

## Abstract

Saffron (*Crocus sativus* L.) is one of the most expensive spices in the world, boasting rich medicinal and edible value. However, the effective development of active natural substances in saffron is still limited. Currently, there is a lack of comprehensive studies on the saffron stigma protein, and the main effect peptides have not been identified. In this study, the total protein composition of saffron stigmas was analyzed, and two main antioxidant peptides (DGGSDYLGK and VDPYFNK) were identified, which showed high antioxidant activity. Then, the stability of two peptides was further evaluated. Furthermore, our results suggested that these two peptides may protect HepG2 cells from H_2_O_2_-induced oxidative damage by significantly improving the activity of endogenous antioxidant enzymes and reducing the malondialdehyde (MDA) content. Collectively, we identified two peptides screened from the saffron protein possessing good antioxidant activity and stability, making them promising candidates for use as functional foods, etc., for health promotion. Our findings indicated that proteomic analysis together with peptide identification is a good method for exploitation and utilization of spice plants.

## 1. Introduction

Saffron (*Crocus sativus* L.), derived primarily from the Mediterranean coast, Central Asia, and some European regions, is also cultivated in regions such as Zhejiang in China [1]. In Chinese herbal literature, it holds significant value as a medicinal herb [2]. Its medicinal part, three-branched red stigma, exhibits physiological activity and medicinal value. Many in vivo studies in animals have indicated a very low or even non-existent toxicity of both saffron and its extracts [3]. In recent years, in-depth studies of saffron found that it is a rich source of multiple active ingredients, including carotenoids, flavonoids, and terpenoids [4]. At present, the most widely studied bioactive compounds in saffron are safranal, crocin, and picrocrocin [5]. Safranal (a monoterpene aldehyde) is the major volatile component of saffron that is responsible for its unique odor [6]; crocin, a water-soluble carotenoid, is the source of its color [7], and picrocrocin is the source of the bitterness [8], as well as anti-inflammatory activity [9], antidepressant properties [10], protection of the cardiovascular system [11], and improving memory [12]. In addition, saffron contains a variety of proteins, polysaccharides, amino acids, organic acids, trace elements, and other ingredients. Currently, the demand for saffron in the domestic and international markets is gradually increasing. However, due to challenges in cultivation and limited yield, it is costly to directly use its natural bioactive substances. Therefore, it is necessary to identify its main active ingredients.

The excessive free radicals generated in the oxidation process pose significant threats to human health. As these radicals contain unpaired electrons, they are highly unstable and tend to take electrons from nearby molecules, including fats, proteins, and DNA, to maintain stability. If the electron of the cell protein molecule is taken away, the protein will be alkylated by the branched chain, forming a distorted molecule, and causing cancer [13]. In addition, free radicals are associated with many diseases, such as inflammation and atherosclerosis, among others. Reducing the level of free radicals in the body by using antioxidant active substances helps to reduce oxidative damage, which is conducive to the delay of aging and the prevention and treatment of related diseases. Therefore, extensive research has been devoted to exploring diverse forms of antioxidants, with accumulating evidence suggesting that natural extracts often possess potent antioxidant properties, emerging as a promising source of novel antioxidants. Antioxidant peptides have been obtained from the enzymatic hydrolysates of food proteins, including milk protein [14], fish protein [15], and walnut [16]. Considering their inherent safety and accessibility, these antioxidant peptides have garnered considerable attention as a focal point in research.

Previous studies have shown that many components of saffron contain good antioxidant properties. The carotenoids of saffron can inhibit the accumulation of intracellular ROS and Ca^2+^ overload [17]. The flavonoids and phenolic compounds in the methanol extract of saffron have good free radical scavenging and iron-reducing power activity [3]. Zhang et al. reported the relations between polysaccharides in saffron and antioxidant capacity, and the best correlation was found with diphenyl-2-picrylhydrazyl (DPPH; 0.996) [18]. Moreover, the hydroethanolic and aqueous extracts of saffron stigmas have great antioxidant activity and can block the reactive oxygen species and intracellular signals’ activation to downregulate the apoptotic pathway, ultimately enhancing cell viability [19]. These scientific reports suggest that saffron is a rich source of natural antioxidants. However, the main functional substances of these extracts remain unknown, and there are few studies on saffron protein.

In the study of saffron protein, Paredi et al. employed proteomic techniques to characterize saffron’s protein composition. By analyzing these protein characteristics, they detected adulteration in saffron [20]. Chen et al. compared the protein profiles of saffron’s flowering and non-flowering groups to identify key proteins that influence the initiation of saffron flowers under cold stress conditions [21]. In addition, other scholars have studied the active peptides in the saffron stigma. In this study, saffron stigma protein was extracted, and the composition of saffron protein was analyzed by HPLC, LC-MS/MS, SDS-PAGE, and Native-PAGE. By comparing these data with existing databases, the main components of antioxidant peptides were screened. Simultaneously, in vitro antioxidant experiments were carried out to verify the antioxidant capacity of these peptides. This research is crucial for understanding the antioxidant activity of saffron and provides novel insights into the activity of saffron peptides.

## 2. Materials and Methods

### 2.1. Total Protein Extraction of Saffron

Extraction of protein from saffron stigmas was carried out according to the method described by Habiba et al., with some modifications [22]. Twenty fresh saffron stigmas were provided by the Laboratory of Resources Protection and Innovation of Traditional Chinese Medicine, Zhejiang A&F University (Hangzhou, China) in October 2022, and their identity was confirmed by Professor Qingsong Shao at Zhejiang A&F University (Hangzhou, China). The saffron stigmas were freeze-dried for 72 h at 0 °C, and ground into powder and stored at −20 °C for subsequent experiments. Then, 0.1 g of dried saffron powder and 8 mL of PBS (pH = 7.0) were fully mixed and placed in a warm water bath at 40 °C for 3 h. The supernatant was obtained by centrifugation at 2000× *g* at 4 °C for 20 min, repeated twice. The supernatant was filtered with a 0.22 μm microporous membrane (ANPEL, Shanghai, China) to remove impurities and obtain a relatively pure saffron protein. Subsequently, the supernatant was fully mixed with cold acetone 1:1. The mixture was centrifuged at 2000× *g* at 4 °C for 20 min, and the precipitate was retained. Finally, it was redissolved with 600 μL of PBS buffer, and the total protein content was determined using the Pierce BCA Protein Assay Kit (Thermo Scientific, Waltham, MA, USA).

### 2.2. Determination of the Whole Protein of Saffron by SDS-PAGE and Native-PAGE

The SDS-PAGE denatured acrylamide gel rapid preparation kit (Sangong Biotech, Shanghai, China) and Native-PAGE denatured acrylamide gel rapid preparation kit (Sangong Biotech, Shanghai, China) were used to prepare 12% separation gel and 5% staking gel. For SDS-PAGE, the sample was fully mixed with 10 μL of 5× denaturing reducing protein buffer (Sangong Biotech, Shanghai, China) and placed in a metal bath at 95 °C for 5 min. For Native-PAGE, the sample was fully mixed with 10 μL of 5× non-denatured reducing protein buffer (Sangong Biotech, Shanghai, China). Then, 40 μL of sample was loaded on each channel for electrophoresis, and the voltage was adjusted to 180 V, then stained with 0.1% Coomassie Brilliant blue (R-250; Sangong Biotech, Shanghai, China) for 2 h. Finally, a decolorization solution (methanol:acetic acid:water = 20:30:350) was used for 8 h [19,23].

### 2.3. Protein Analysis of Saffron by High-Performance Liquid Chromatography (HPLC)

Protein separation was performed using HPLC according to the method described by Gu et al., with some modifications [24]. The extracted saffron protein was filtered using a 0.22 µm filter membrane and then analyzed by HPLC. An Alliance system (Waters e2695 Separations Module and Waters 2998 PDA Detector) was used for the HPLC and a Sepax GP-C18 column (4.6 × 250 mm, 5 µm, Agilent, Santa Clara, CA, USA) was used as the stationary phase. The HPLC conditions were as follows: mobile phase of methanol (MeOH, HPLC grade) (A) and water (B) under the following elution program: 0–30 min, 10–90% A; 30–35 min, retained at 90% A; 35–36 min, 90–10% A; 36–66 min (the re-equilibration time), retained at 10% A. The flow rate was set at 1.0 mL/min, and the column temperature was maintained at 25 °C. The injection volume was 10 μL, and the detection wavelength was set at 280 nm.

### 2.4. Identification of Saffron’s Functional Peptides by Liquid Chromatography Mass Spectrometry (LC-MS/MS) and Proteome Sequencing

The peptide compounds of the four new antioxidant peptides were identified by LC-MS/MS according to Fan et al., with minor modifications [25]. Saffron protein was analyzed by label-free tandem mass tag, followed by trypsin hydrolysis (mass ratio 1:50) at 37 °C for 20 h. LC-MS (nano-LC-QE; Thermo Fisher) was used for analysis. Qualitative information on the target protein molecules was obtained by analyzing the LC-MS/MS data using MASCOT Sever 2.7 and other mass-matching software. The tryptic digest was chromatographed, and the eluted solution was composed of A (0.1% formic acid aqueous solution) and B (0.1% formic acid aqueous solution of acetonitrile (84% acetonitrile)). After the chromatographic column was balanced with 95% A, the protein of saffron was loaded into the Trap by an automatic sampler. Mass-to-charge ratios of peptides and peptide fragments were collected, and 20 fragment maps (MS 2 scan) were collected after each full scan. The mass spectra were analyzed by Proteome Discoverer 1.4 software (Thermo Scientific, MA, USA), and the potential main antioxidant peptides in saffron were screened out by comparing with the BIOPEP database.

### 2.5. In Vitro Antioxidant Acrivity

#### 2.5.1. DPPH Radical Scavenging Activity

The DPPH radical scavenging activity analysis was carried out using the method of Dong et al., with slight changes [26]. Here, 380 μL of 0.1 mmol/L DPPH solution and 20 μL sample extracts were mixed in the test tube. The mixture was kept at room temperature (25 °C) in darkness for 20 min. Equal volumes (380 μL) of DPPH solution and ethanol were mixed as a control. The absorbance of each reaction mixture was measured at 515 nm with a 200 μL reaction mixture using a microplate spectrophotometer. The DPPH radical scavenging capacity was calculated using the following equation:DPPH radical scavenging activity (%) = (A − A_0_)/A × 100
where A is the absorbance of DPPH and ethanol, and A_0_ is the absorbance of DPPH, ethanol, and sample. The experiment was repeated three times.

#### 2.5.2. 2,2′-Azino-bis-(3-ethylbenzthiazoline-6-sulfonic acid) (ABTS) Radical Scavenging Activity Assay

A modified ABTS radical scavenging activity (ARSA) assay was used [27]. The ABTS radical cation stock solution contained 7 mM of ABTS and 2.45 mM of potassium persulfate in 10 mM of phosphate buffer and was stored at room temperature for 20 h in the dark. The working solution was prepared by diluting the ABTS radical cation stock solution in phosphate buffer with an absorbance of 0.70 ± 0.02 at 734 nm. Different concentrations of samples (10 μL) were mixed with 190 μL of ABTS working solution and stored at room temperature for 20 min in the dark. Then, the absorbance was measured at 734 nm, and the scavenging rate was calculated according to the following equation:SA(%) = (A − A_0_)/A × 100
where A is the absorbance of the blank and A_0_ is the absorbance of the sample.

#### 2.5.3. Ferric-Reducing Antioxidant Power (FRAP) Assay

The FRAP reagent was prepared by mixing 1 mL of 10 mM TPTZ solution in 40 mM of HCl, 1 mL of 20 mM FeSO_4_ × 7 H_2_O, and 7 mL of 0.3 M acetate buffer (pH 3.6) [28]. We added 18 μL of distilled water to the different 6 µL concentration samples, which were then mixed with 180 µL of FRAP reagent and stored at room temperature for 10 min. Then, the absorbance was measured at 593 nm. The standard curve was established using the FeSO_4_ standard solution with a linear equation: ΔA_593_ = 0.0862C + 0.0001 (R^2^ = 0.9996), where ΔA_593_ is the absorbance difference between the absorbance of the sample at 593 nm and the absorbance of the blank at 593 nm, and C is the final Fe^2+^ concentration.

### 2.6. H_2_O_2_ Induced Oxidative Damage in a HepG2 Cell Model Experiment

To appraise the effects of saffron stigma peptides on the oxidative stress of HepG2 cells induced by H_2_O_2_, the cell viability was determined on the basis of the method described by Wen et al. [29]. The HepG2 cells were cultured in DMEM (Gibco, New York, NY, USA) containing 10% (*v*/*v*) fetal bovine serum (HyClone, Logan, UT, USA) and 1% (*v*/*v*) penicillin/streptomycin (Gibco, New York, NY, USA) at 37 °C and 5% CO_2_.

The cells were passaged every 2 days, with a passage ratio of 1:2 when the cells grew to more than 80% confluence. Then, the HepG2 cells were distributed in 96-well culture plates at a density of 2.5 × 10^4^ cells per well. When the cells were cultured for 24 h in DMEM medium, the medium was removed and replaced with a new medium containing various concentrations of H_2_O_2_ (100 mM, 300 mM, 500 mM, 700 mM, 900 mM, and 1100 mM), and the control cells were grown in DMEM. MTT (3-(4,5-dimethylthiazol2-yl)-2,5-diphenyltetrazolium bromide; Sigma-Aldrich, St. Louis, MO, USA) was added to the wells and the cells were incubated for 2 h at 37 °C. Then, DMSO was added to the wells for dissolution of formazan granules produced by the live cells and the absorbance measurements were performed at 540 nm to ensure the half inhibitory concentration (IC_50_) of H_2_O_2_ [30].

The safe and harmless peptide concentrations were also determined by the above method. Different concentrations of peptides (0.01 mg/mL, 0.02 mg/mL, 0.04 mg/mL, 0.06 mg/mL, 0.08 mg/mL, and 0.1 mg/mL) also act on cells, and cell viability was detected by MTT to screen out the safe and harmless peptide concentrations.

### 2.7. Intracellular Antioxidant Activity Assay

The protective effects of saffron stigma peptides against H_2_O_2_-induced cytotoxicity were determined to investigate the intracellular antioxidant activity. Briefly, HepG2 cells were seeded in 6-well plates at a density of 3 × 10^6^ cells/mL. After incubation for 24 h, the normal cultured cells were used as a control group. The DMEM containing H_2_O_2_ (IC_50_) was a model group. Peptide groups included H_2_O_2_ (IC_50_) and different concentrations of peptides mixed in the DMEM. After that, the SOD, CAT, MDA, and GSH-Px were detected according to different assay kits (Sangong Biotech, Shanghai, China) [31,32].

### 2.8. Determination of Stability of Saffron Antioxidant Peptides

The pH stability and thermostability of saffron stigma peptides were analyzed in accordance with the previous method, with slight modification [33]. The effects of temperature and pH on the stability of antioxidant peptides were detected. Antioxidant peptides (1 mg/mL) were placed in 25 °C, 37 °C, 60 °C, 80 °C, and 100 °C water baths, respectively. Then, the DPPH free radical scavenging rate and reducing antioxidant capacity were detected. The pH values were adjusted to 2.0, 4.0, 6.0, 7.0, 8.0, and 10.0, and the solution was placed in a 37 °C water bath for 1 h. After the end of the incubation period, the pH value of the solution was adjusted back to 7.0, and the DPPH free radical scavenging rate and reducing antioxidant capacity were measured.

### 2.9. Statistical Analysis

Each experiment was repeated at least three times in parallel, and data were expressed as the means and standard deviations of three experiments. Error bars in the figures represent standard deviation. Using the SPSS 26 statistical software, the data were analyzed using a two-way analysis of variance (ANOVA; Chicago, IL, USA), followed by the least significant difference test at a 95% confidence level. To indicate statistically significant differences, different letters or signs above the columns were used.

## 3. Results

### 3.1. The Composition and Antioxidant Activity of Total Stigma Protein

Figure 1A shows photographs displaying the overall shape of saffron and stigmas. To explore the molecular weight and analyze the subunits of the protein, SDS-PAGE electrophoresis was conducted [34]. The total stigma protein content of saffron was determined by SDS-PAGE, Native-PAGE, and HPLC. As shown in Figure 1B, SDS-PAGE showed that the bands of stigma proteins were mostly in the 35 to 50 kDa range. Our results are consistent with Paredi et al.’s research using SDS-PAGE on Spanish saffron [20]. However, as shown by the Native-PAGE results, it can be speculated that proteins exist in the form of polymers in the natural state. Protein hydrolysate is a complex mixture of partially unhydrolyzed protein, polypeptides with different chain lengths, hydrophobicity, and net charge, as well as free amino acids. Next, the HPLC method was used to analyze the intact protein and enzymatic protein of saffron. In the unhydrolyzed samples, 17 peptides were detected, while after 16 h of enzymolysis, only 11 peptides were detected (Figure 1D). It is evident that in comparison to the non-enzymatic hydrolysis condition, the detected protein peak area decreased.

Subsequently, the ABTS method was utilized to determine the total antioxidant activity of the stigma protein. The stigma protein exhibited strong ABTS radical scavenging activity (Figure 1C). The IC_50_ value of the stigma protein was 1.89 mg/mL, while protein hydrolysates of canola against ABTS activity showed lower IC_50_ values, which ranged around 12–85 µg/mL [35]. Therefore, the obtained stigma protein had a scavenging activity stronger than canola protein hydrolysates. These results demonstrated the presence of bioactive peptides in the stigma protein, representing an additional value for their incorporation into functional foods.

### 3.2. Identification of Peptide Sequences

Saffron stigma protein was digested with trypsin and analyzed by LC-MS/MS (nano-LC-QE). The analyte was converted into bandgap ions in the ion source, and an ion beam was formed by accelerating the electric field and entering the mass spectrometer. LC-MS analysis resulted in the detection of a total of 2311 proteins with 12,237 peptides (Appendix A). The BIOPEP database was used to evaluate the potential functional proteins in saffron protein, and 39 functional proteins were found. The ten proteins with the highest relative expression levels were analyzed for functional information (Table 1).

Wu et al. declared that specific amino acids and different amino acid sequences might affect the structural and physicochemical properties of peptides, thereby altering their antioxidant activity [36]. We screened 289 saffron antioxidant peptides by comparing with the BIOPEP database. Among them, the saffron peptides with high abundance in the range from tetrapeptide to decapeptide were selected. As shown in Figure 2A, saffron peptides with antioxidant function were screened, and the selected peptides basically had hydrophobic and aromatic amino acids. It has been reported that hydrophobic amino acids can supply electrons, while aromatic amino acids can supply protons to electron-deficient substances, ultimately enhancing the antioxidant capacity [37,38]. Among them, the expression levels of three peptides, namely DMSQADFGR, DGGSDYLGK, and VDPYFNK, were the highest. The sequences ADF, YLG, and PYFNK have been previously reported to exhibit good antioxidant activity. ADF, isolated from the proteolytic products of the bean residue, showed a certain antioxidant activity [39]. Amigo et al. found that YLG has good antioxidant activity [40]. PYFNK peptides showed good ABTS scavenging activity and DPPH radical scavenging activity [41]. However, Peña-Ramos et al. pointed out that the ability of Ser, Ala, and Asp to scavenge OH was limited [42]. Clausen et al. reported that Phe had a low ability to scavenge ABTS and showed scavenging activity when the concentration reached 10^−3^ mol/L [43]. Therefore, to summarize the above reports, the DGGSDYLGK and VDPYFNK were finally screened, and the structure of the two peptides can be seen in Figure 2B,C.

### 3.3. Analysis of Antioxidant Capacity of Saffron Stigma Active Peptides

In this study, we used three different methods to evaluate the antioxidant capacity of saffron protein. The two peptides were, respectively, synthesized and then tested for activity (each 4 mg, 85% purity, GenScript, Jiangsu, China). The ABTS values of DGGSDYLGK and VDPYFNK peptides were significantly improved in the concentration range of 0.1 mg/mL to 1.0 mg/mL, with IC_50_ values of 0.2930 mg/mL and 0.1657 mg/mL, respectively. These values were significantly higher than those of jackfruit seed [44]. The results indicated that DGGSDYLGK and VDPYFNK peptides have good ABTS free radical scavenging activity and have the potential to act as natural antioxidants. They may be the main peptides responsible for the antioxidant effect of saffron peptides. The DPPH values of saffron peptides are shown in Figure 3A,B. The IC_50_ value of DGGSDYLGK was 0.3901 mg/mL, and the IC_50_ value of VDPYFNK was 0.6411 mg/mL. The values are higher than the published values of the lotus seed [45] and sesame protein hydrolysate [46]. The antioxidant activity of PYFNK and YLG has been reported previously [40,41], and we used these two oligopeptide sequences as a reference to screen saffron polypeptides. According to our results, the IC_50_ values of ABTS and DPPH of VDPYFNK were much higher than those of PYFNK. Compared with the single YLG antioxidant activity experiment performed by previous researchers, we used three methods, which were more complete and systematic.

The FRAP of DGGSDYLGK peptides to Fe^3+^ ranged from an average of 0.144 μmol/mL to 0.0549 μmol/mL, in a dose-dependent manner. The FRAP of VDPYFNK peptides to Fe^3+^ ranged from an average of 0.167 μmol/mL to 0.063 μmol/mL, in a dose-dependent manner. Figure 3A,B show that the FRAP was greater than the antioxidant activity of the peptides in Ganoderma lucidum [47].

### 3.4. Effect of Saffron Peptides on HepG2 Cells Injured with H_2_O_2_

H_2_O_2_ can penetrate the cell membrane and lead to oxidative stress. The intercellular antioxidant activities of DGGSDYLGK and VDPYFNK were investigated in HepG2 cells. Before the cellular antioxidant activity assay, the effects of these different concentrations of peptides and the IC_50_ of H_2_O_2_ on the HepG2 cells were investigated using the MTT assay. As shown in Figure 4A,B, the peptides DGGSDYLGK and VDPYFNK had no significant effects on the viability of HepG2 cells in the range of 0.01–0.1 mg/mL. Therefore, the tested dose range (0.01–0.08 mg/mL) was chosen for further study. In Figure 4C, the H_2_O_2_ content of 300 mM could make the cell viability decline by about 50%. Therefore, HepG2 cells were pretreated with saffron peptides (0.08, 0.04, 0.02, and 0.01 mg/mL) for 12 h, and then injured with 300 mM of H_2_O_2_ for 4 h. To further clarify the protective mechanism of saffron peptides on H_2_O_2_-mediated oxidative stress injury to HepG2 cells, SOD, CAT, GSH, and MAD activities in HepG2 cells were investigated.

The SOD could make O^2−^ transform to H_2_O_2_ and CAT could decompose H_2_O_2_ to H_2_O and O_2_, eliminating the ROS. Therefore, these antioxidant enzymes can inhibit the accumulation of oxygen-free radicals in the body and protect cells from oxidative damage [48]. The thiol group of cysteine in GSH-Px is a potent reducing agent that can perform several important cellular functions, such as metabolism, catalysis, and transport. Therefore, GSH-Px is also one of the indicators that can reflect the degree of oxidative damage of cells [49]. The lipid peroxide generated from the reaction between ROS and the double bond of polyunsaturated fatty acids can cause the degradation of the cell membrane. Then, a series of aldehyde compounds will be set free, and MDA is one of them. Therefore, the content of MDA can indirectly reflect the ROS-mediated cell oxidative stress damage degree [50,51].

Figure 4H–M show that the SOD, CAT, and GSH-Px activities in HepG2 were all significantly decreased after being exposed to 300 mmol/L of H_2_O_2_ for 4 h, and the MDA content level increased. These results revealed that H_2_O_2_ destroyed the dynamic balance of cells and led to cells being damaged by oxidative stress. Nevertheless, when pretreated by DGGSDYLGK and VDPYFNK at different concentrations for 12 h, the activity of SOD, CAT, and GSH-Px were all higher than those in the H_2_O_2_ damaged group (*p* < 0.05). In an earlier study, researchers found that hemp seed protein hydrolysates (HPH) at a concentration of 0.4 mg/mL enhanced the activities of SOD (2.6-fold), CAT (2.5-fold), and GSH-Px (2.1-fold) in H_2_O_2_-treated HepG2 cells, compared with the H_2_O_2_ damaged group [33]. In our study, compared with the H_2_O_2_-treated group, the activities of SOD, CAT, and GSH-Px in HepG2 cells pre-incubated with the DGGSDYLGK concentration of 0.4 mg/mL increased up to 5.6-fold, 2.1-fold, and 2.4-fold, respectively, and with the VDPYFNK concentration of 0.4 mg/mL increased up to 4.9-fold, 2.5-fold, and 1.6-fold, respectively.

In our study, the activities of MAD were lower than the H_2_O_2_ damaged group, and a higher content of MDA reflected a higher ROS level and more severe cell oxidative stress [52]. From Figure 4N,O, compared with the model group, MDA decreased by 4-fold at a 0.08 concentration mg/mL of DGGSDYLGK (*p* < 0.001), and decreased by 1.6-fold at a 0.08 concentration of VDPYFNK (*p* < 0.05). The MDA and GSH-Px contents of the two peptides at a concentration of 0.02 mg/mL were significantly higher than the reported antioxidant peptides of wild jujube seed [32]. Taken together, these results suggested that the DGGSDYLGK and VDPYFNK could protect HepG2 cells against H_2_O_2_-stimulated oxidative damage by enhancing the activities of antioxidant enzymes and reducing the MDA content.

### 3.5. The Stability of Saffron Antioxidant Peptides

Previous studies have reported on how pH and temperature could affect the peptides’ antioxidant activity [53]. The pH and temperature stability are important, as these stability profiles will determine the optimal storage conditions for antioxidant peptides. Similarly, the stability of these peptides in food systems with a particular acidity is also of interest, as it can affect their functionality and bioavailability [15,54]. Therefore, it is essential to consider the pH and temperature stability profiles of antioxidant peptides when evaluating their potential for use in food processing or as functional additives in food systems.

To assess how the DGGSDYLGK and VDPYFNK would be affected by different temperature and pH conditions, the range of the temperature test was 25 to 100 °C, while the range of the pH test was 2 to 10. Figure 5A,C show that the DPPH and FRAP gradually decreased with the increase in temperature when the DGGSDYLGK and VDPYFNK solutions were in the range of 25 to 100 °C. This is consistent with the research by Zhu et al., showing that the DPPH of the antioxidant peptide derived from Jinhua ham sharply decreased after incubation from 60 °C to 100 °C [55]. Similarly, Zhang et al. reported that the activity of antioxidant peptides in Antarctic krill decreased with the increase in temperature [34].

The peak areas of the two peptides gradually decreased with the increase in temperature (Figure 5B,D and Table 2). This suggests that the structure of the peptides may be destroyed at high temperatures, which could affect their stability and processing properties. However, it is noteworthy that even under conditions where the temperature exceeded 60 °C, the antioxidant activity of the saffron peptide remained at more than 50% of its original activity. This indicates that saffron peptides have good thermal stability and can retain their antioxidant activity even under high-temperature conditions.

As seen in Figure 5E,G, within the pH range of 2.0 to 10.0 for saffron peptide solutions, the FRAP value initially increased and then decreased with increasing pH. Notably, at pH 6.0 and pH 8.0, DGGSDYLGK exhibited strong scavenging activities against both FRAP and DPPH, while at pH 8.0, VDPYFNK showed potent scavenging effects on FRAP and DPPH. Additionally, Figure 5F,H reveal that VDPYFNK’s peak height significantly varied at pH 2.0, whereas DGGSDYLGK’s peak height remained relatively stable. In Table 2, DGGSDYLGK had the smallest peak area at pH 2.0. This means that acidic environments can affect peptides’ solubility, impeding their binding with free radicals and reducing their antioxidant activity [56].

All in all, this means that the two peptides can maintain good antioxidant activity under pH 8.0 conditions.

## 4. Conclusions

In this study, the total protein composition of saffron stigmas was analyzed, the peptide spectrum was characterized, and the antioxidant activity of the total protein was detected. The most important finding was the two peptides (DGGSDYLGK and VDPYFNK), where the main antioxidant active ingredients in the stigma protein were screened out, and these two peptides exhibited excellent antioxidant activities both extracellularly and intracellularly. The discovery of DGGSDYLGK and VDPYFNK may benefit the effective development of active natural substances and provide valuable insights for the development of novel antioxidant peptide-based drugs or functional foods.

## Figures and Tables

**Figure 1 antioxidants-13-00378-f001:**
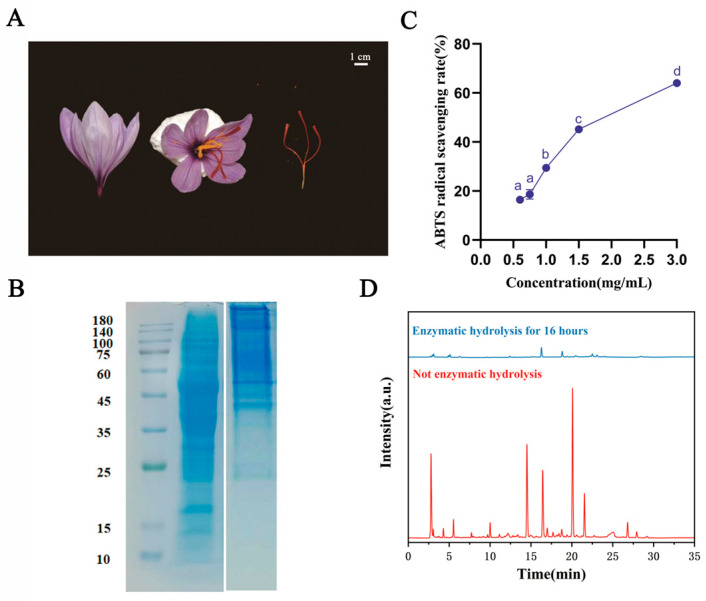
Saffron stigma protein composition and antioxidant activity characteristics. (**A**) Photographs displaying the overall shape of saffron and stigma. (**B**) SDS-PAGE and Native-PAGE of saffron stigma protein. (**C**) The ABTS radical scavenging capacity of total stigma protein. (**D**) Chromatographic diagram of saffron stigma protein before and after enzymatic hydrolysis at 280 nm. Reported values show means and standard deviations over three biological replicates, and ^a–d^ values with the same letters indicate no significant difference for the different concentrations (*p* < 0.05).

**Figure 2 antioxidants-13-00378-f002:**
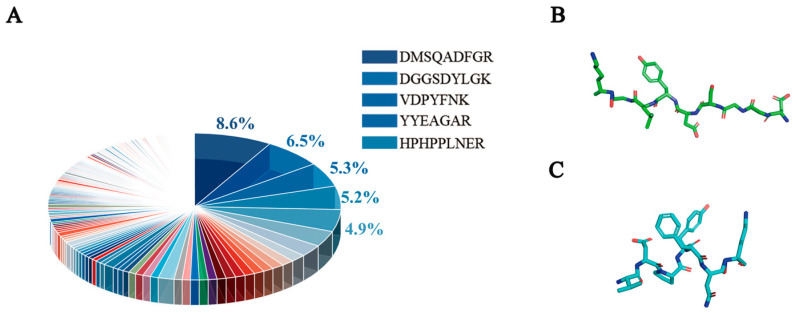
Saffron peptides with antioxidant function in the range of four to ten peptides. (**A**) The richness of antioxidant peptides produced by enzymatic hydrolysis of saffron stigma protein. (**B**,**C**) The simulated structures of DGGSDYLGK (**B**) and VDPYFNK (**C**).

**Figure 3 antioxidants-13-00378-f003:**
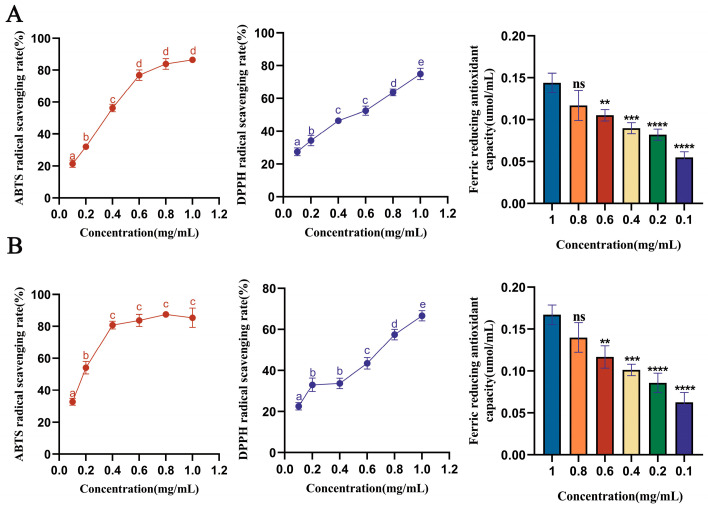
Determination of the antioxidant activities of stigma peptides. (**A**) ABTS, DPPH, and FRAP of DGGSDYLGK. (**B**) ABTS, DPPH, and FRAP of VDPYFNK. Reported values show means and standard deviations over three biological replicates. ^a–e^ Values with the same letters indicate no significant difference for the different concentrations (*p* < 0.05): ** *p* < 0.01, **** p* < 0.001, and **** *p* < 0.0001, different synthetic peptide concentration groups vs. the model group. Using the SPSS statistical software, the data were analyzed using a two-way ANOVA.

**Figure 4 antioxidants-13-00378-f004:**
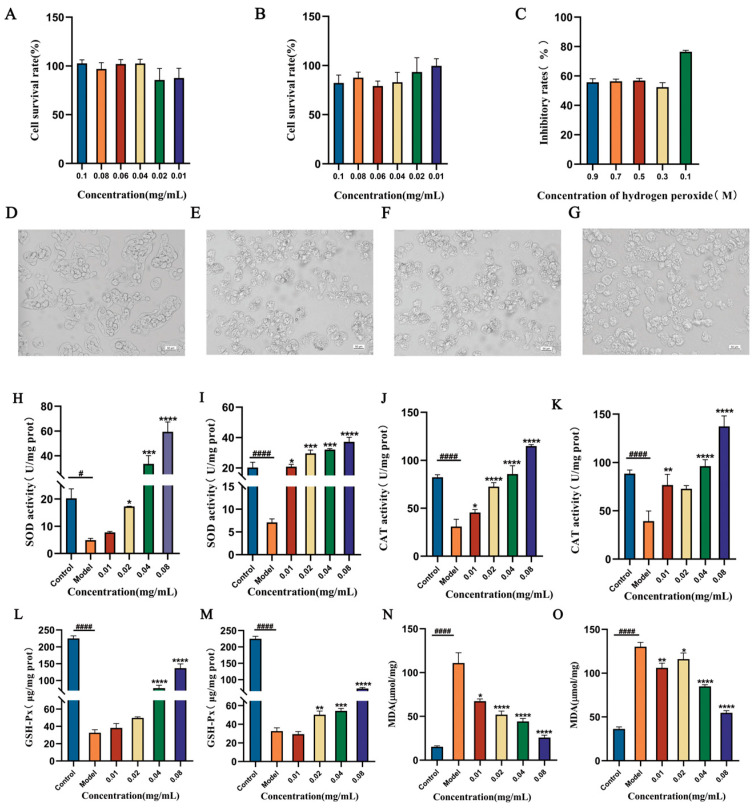
The SOD, CAT, GSH-Px, and MDA activities and the survival rates of HepG2 cells under different treatments. (**A**,**B**) The effects of DGGSDYLGK (**A**) and VDPYFNK (**B**) on the cell survival rate. (**C**) H_2_O_2_ acting on cells. (**D**) HepG2 cells in a normal state. (**E**–**G**) HepG2 cells in 0.5 M, 0.3 M, and 0.1 M H_2_O_2_-induced oxidative damage states. (**H**,**I**) The SOD activities after adding DGGSDYLGK (**H**) and VDPYFNK (**I**) to oxidative damage cells. (**J**,**K**) The CAT activities after adding DGGSDYLGK (**J**) and VDPYFNK (**K**) to oxidative damage cells. (**L**,**M**) The GSH-Px activities after adding DGGSDYLGK (**L**) and VDPYFNK (**M**) to oxidative damage cells. (**N**,**O**) The MDA activities after adding DGGSDYLGK (**N**) and VDPYFNK (**O**) to oxidative damage cells. Reported values show means and standard deviations over three biological replicates: ^#^
*p* < 0.05 and ^####^
*p* < 0.0001, control group vs. model group; * *p* < 0.05, ** *p* < 0.01, *** *p* < 0.001, and **** *p* < 0.0001, different synthetic peptide concentration groups vs. the model group. Using the SPSS statistical software, the data were analyzed using a two-way ANOVA. Ratio scale = 50 μm.

**Figure 5 antioxidants-13-00378-f005:**
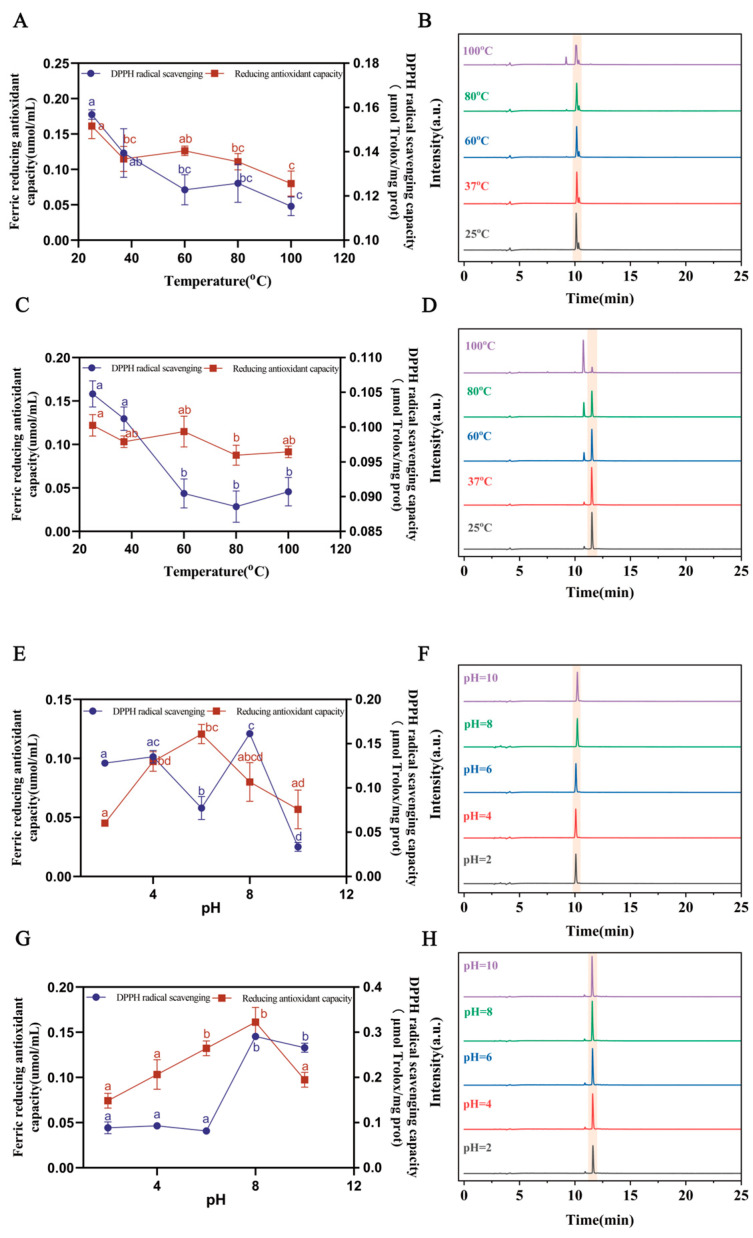
DPPH scavenging activity and FRAP of DGGSDYLGK and VDPYFNK subjected to different temperature and pH treatments. (**A**,**B**) The FRAP and DPPH (**A**) and chromatographic diagram (**B**) of DGGSDYLGK at different temperatures. (**C**,**D**) The FRAP and DPPH (**C**) and chromatographic diagram (**D**) of VDPYFNK at different temperatures. (**E**,**F**) The FRAP and DPPH (**E**) and chromatographic diagram (**F**) of DGGSDYLGK at different pH values. (**G**,**H**) The FRAP and DPPH (**G**) and chromatographic diagram (**H**) of VDPYFNK at different pH values. Reported values show means and standard deviations over three biological replicates. ^a–d^ Values with the same letters indicate no significant difference for the different temperatures (*p* < 0.05).

**Table 1 antioxidants-13-00378-t001:** The 10 proteins with the highest relative expression in the saffron proteome.

Accession	Amino Acids	Molecular Weight (kDa)	Abundance	Annotation
D2T0A5	91	8.8	2173002816	Non-specific lipid-transfer protein (fragment)
A0A5J6ANU7	497	53.4	1602808441	Aldehyde dehydrogenase
A0A2U8ZTY0	537	58.3	1467403203	Aldehyde dehydrogenase 2B4
A0A6H0C818	462	51.2	1347643392	Glycosyltransferase
A0A075M6P3	477	53.2	818608184.3	Glycosyltransferase
A0A1S5T4X6	240	26.7	485796513.8	SOUL heme-binding protein
A0A3G1GZP7	537	58.4	360497340	Aldehyde dehydrogenase
A0A5J6ANM0	504	55.1	347338259	Aldehyde dehydrogenase
A0A1S5VK40	507	57.3	335751151	Beta-glucosidase 12
A0A075M6K1	434	48	262639198	Glycosyltransferase

**Table 2 antioxidants-13-00378-t002:** The peak areas of DGGSDYLGK and VDPYFNK under different temperatures and pH treatments.

Peptide	Temperature (°C)					pH				
	25	37	60	80	100	2	4	6	8	10
DGGSDYLGK	0.08612	0.07396	0.07549	0.07357	0.074	0.09748	0.13064	0.12197	0.13675	0.1374
VDPYFNK	0.13439	0.13967	0.11136	0.18938	0.01905	0.09583	0.09639	0.09556	0.09276	0.09625

## Data Availability

Data are contained within the article.

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
