# Peer review of "Identification and Functional Validation of Two Novel Antioxidant Peptides in Saffron"

_antioxidants, 2024, doi:10.3390/antiox13030378_

Round 1

Reviewer 1 Report

The paper is interesting, since it deals with characterization of two peptides aimed with antioxidant activity.

The peptides have already been identified, but, at my knowledge, their activity were not reported in literature.  The manuscript is done quite well and the experiments are congruent with the aim of the work, but a deep revision of the manuscript is need before publication.

SPECIFIC POINTS

Title: I think that the title can be simplified by removing the last sentence. “the exploitation and utilization of spices plants” is not part of the manuscript and it can be removed.

Abstract. the text is unclear. The 17-19 lines are quite incomprehensible. A rewrite of the abstract is strongly suggested.

Introduction: It seems written in a very hasty and superficial manner.

In particular: line 39-42 deal with the main component of saffron. They are important for their chemical structure, but any class of molecules are taken into account; for example, what are  Safranal, crocin, and picrocrocin? a brief description would be added.

Line 48-50: a general definition of the free radical activity is not necessary, but a greater precision when describing their effect would be appreciated

Line 68-74: What is the point of reporting exactly the antioxidant activity of various extracts? It would be preferable to cite some results describing the bioactivity of total extract, polysaccharides, polyphenols an so on, and then to introduce the theme of the proteins and peptides.

Line 85: what does it mean?

Finally, the manuscript deals with bioactive peptides, but any reference to the potential of these peptides is reported. Please, add some consideration about this.

METHODS: they have to be revised in some points

line 107. bovine serum albumin kit? Perhaps, Comassie kit? Check it.

line 112: 5% concentrated gel. Perhaps, staking gel….

general consideration: the 2.5.1/.2/.3 paragraphs can be merged as “in vitro antioxidant acrivity”

2.7 paragraph: is it the final step of the 2.6  paragraph. Why are they in  different paragraph?  Firstly, the optimal concentration  is determined (2.6); after the  effect of 0.5 mg/mL peptides on cell viability is determined. Is it right?

RESULTS:

line 234: hydrolysate? The comparison of  native and SDS-PAGE indicates  the presence of polymers and/or  aggregates in the PBS extract.

line 238-240: Please rewrite the result; it is not understandable

line 245. a comparison between stigma and canola peptides is reported. When are canola peptides taken into account?

line 247: in the text  three tripeptides are reported, but there are not further mentioned. Why are these peptides cited?

Figure 2: the pie chart is not clear. what are the different colors meaning for?

General consideration: it is not clear if the amount of the required peptides has been obtained from purification or, after the identification they have been synthesized? How many mg of peptides are need for the experiments?

the 3.4 and 3.5 paragraphs are more understandable, but, in the light of  previous observations, a revision of the language would be desirable

Figure 2 (see the note in the comment to result reported above)

Author Response

Thank you. Please refer to the attachment for our response. Thank you for taking the time to process our manuscript. Your effort and attention are greatly appreciated.

Reviewer 2 Report

The manuscript is interesting, very well structured, interesting results, a lot of methods are done and well presented, thoroughly written, has a significant scientific contribution, the English is comprehensible.

Dear authors, I reviewed in detail the paper entitled "Identification and functional validation of two novel antioxidant peptides in saffron: An example of the exploitation and utilization of spices plants". 

These are my comments and suggestions.

Lines 14 and 31: Crocus sativus L., L. should not be in italic

In Figure 1C, Figure 3 and Figure 4, concentrations should be written as mg/mL, not mg/ml.

Line 254: a-d values, not a-b values.

Line 306: Below Figure 3, check a-bvalues with same letters indicate no significant difference of different concentration…, I think it should be written a-evalues…

Write in Materials and methods in details which chemicals and reagents were used for each method and their manufacturers and countries of manufacture.

Did you have a positive control for antioxidant activity?

Make sure you write the full name for all abbreviations the first time. After that, use only abbreviations.

Line 190: Sigma-Aldrich (missing the name of city of manufacturer and country).

Line 203: 3*106 cells/mL, number 6 should be in superscript.

I did not notice that FIgure 1A appears anywhere in the manuscript in the results. The figures should be listed in order.

Under Figures 3 and 4, you should also write which test was used for the statistical analysis.

Line 416: a-bvalues with same letters…, check this a-b values under Figure 5.

Match tables 1 and 2 in terms of appearance and borders, and line thicknesses.

Make sure all references are well written and in accordance with the instructions of the journal (now they are not).

The manuscript is interesting, very well structured, interesting results and well presented, thoroughly written, English is understandable. I suggest acceptance after accepting all comments. I suggest acceptance after minor revision.

Kind regards

Author Response

(The authors gave the same response as above.)

Reviewer 3 Report

1.     Structural identification is critical for the discovery of bioactive ingredients, and some important data are missing regarding the characterizations of the two main peptides such as Proton and Carbon NMR data, 2DNMR, UV-vis spectrometry, and FT-IR, which are highly suggested to be supplemented. Additionally, information on yield and the purity is needed.

2.     In plant materials, some data are missing and recommended to be added, such as a picture of the plant, especially during the collection process, information about who collected and identified the plant, collection time, data regarding the place of collection, and voucher number of the plant.

3.     The manuscript includes data regarding the identification of peptides by LC-MS/MS, but I couldn't find the LC-MS/MS Chart in the supplementary material, which is highly suggested to be added.

4.     The resolution of picture number 1b is really low, which greatly affects the readability of this manuscript. It is highly suggested that these pictures could be reconstructed.

1.     Structural identification is critical for the discovery of bioactive ingredients, and some important data are missing regarding the characterizations of the two main peptides such as Proton and Carbon NMR data, UV-vis spectrometry, and FT-IR, which are highly suggested to be supplemented. Additionally, information on yield and the purity is needed.

2.     In plant materials, some data are missing and recommended to be added, such as a picture of the plant, especially during the collection process, information about who collected and identified the plant, collection time, data regarding the place of collection, and voucher number of the plant.

3.     The manuscript includes data regarding the identification of peptides by LC-MS/MS, but I couldn't find the LC-MS/MS Chart in the supplementary material, which is highly suggested to be added.

4.     The resolution of picture number 1b is really low, which greatly affects the readability of this manuscript. It is highly suggested that these pictures could be reconstructed.

Author Response

(The authors gave the same response as above.)

Round 2

Reviewer 1 Report

The authors have reported all suggested corrections. The manuscript is now suitable for publication

No comments to add.

Reviewer 3 Report

Thank you for revising the manuscript. In my opinion, it's suitable for publication. Thanks!

Thank you for revising the manuscript. In my opinion, it's suitable for publication. Thanks!